# Identification of biomarkers that predict response to subthalamic nucleus deep brain stimulation in resistant obsessive–compulsive disorder: protocol for an open-label follow-up study

Shyam Sundar Arumugham [1], Dwarakanath Srinivas,[2] Janardhanan C Narayanaswamy,[1] TS Jaisoorya [1], Himani Kashyap,[3] Philippe Domenech,[4,5] Stéphane Palfi,[4,5] Luc Mallet,[6,7] Ganesan Venkatasubramanian,[1] YC Janardhan Reddy[1]

For numbered affiliations see end of article.

**Correspondence to**
Dr Shyam Sundar Arumugham;
a.shyamsundar@gmail.com

## ABSTRACT

**Introduction** Deep brain stimulation (DBS) of bilateral anteromedial subthalamic nucleus (amSTN) has been found to be helpful in a subset of patients with severe, chronic and treatment-refractory obsessive–compulsive disorder (OCD). Biomarkers may aid in patient selection and optimisation of this invasive treatment. In this trial, we intend to evaluate neurocognitive function related to STN and related biosignatures as potential biomarkers for STN DBS in OCD.

**Methods and analysis** Twenty-four subjects with treatment-refractory OCD will undergo open-label STN DBS. Structural/functional imaging, electrophysiological recording and neurocognitive assessment would be performed at baseline. The subjects would undergo a structured clinical assessment for 12 months postsurgery. A group of 24 healthy volunteers and 24 subjects with treatment-refractory OCD who receive treatment as usual would be recruited for comparison of biomarkers and treatment response, respectively. Baseline biomarkers would be evaluated as predictors of clinical response. Neuroadaptive changes would be studied through a reassessment of neurocognitive functioning, imaging and electrophysiological activity post DBS.

**Ethics and dissemination** The protocol has been approved by the National Institute of Mental Health and Neurosciences Ethics Committee. The study findings will be disseminated through peer-reviewed scientific journals and scientific meetings.

### Strengths and limitations of this study

► Multimodal evaluation of biomarkers through neuro-imaging, scalp/invasive electrophysiological recording and neurocognitive evaluation.
► Comparison group of treatment-refractory obsessive–compulsive disorder subjects with sufficient follow-up to help evaluate treatment response to deep brain stimulation.
► Limitations include open-label treatment and un-blinded assessment.

## INTRODUCTION

Obsessive–compulsive disorder (OCD) is a common neuropsychiatric condition with a lifetime prevalence of 2%–3%.[1] It is among the top 10 causes of neuropsychiatric disabilities worldwide.[2] The first-line treatment of OCD includes selective serotonin reuptake inhibitors (SRIs) and/or cognitive behaviour therapy (CBT).[3][4] However, around 20%–30% of patients do not respond to standard treatment strategies.[5] Neurosurgical interventions are considered in patients with chronic, severe and treatment-refractory OCD.[6] Ablative neurosurgical procedures, such as gamma ventral capsulotomy, are helpful in around 45%–65% of patients with treatment-refractory OCD.[5] Due to the potential irreversible and severe adverse effects associated with ablative procedures, there has been a surge of interest in deep brain stimulation (DBS) as an alternate treatment. DBS involves electrical stimulation of the subcortical regions through surgically implanted microelectrodes. High-frequency electrical stimulation through these electrodes modulate the activity of dysfunctional neuronal circuits.[7][8] Unlike ablative procedures, the stimulation in DBS can be modulated to optimise improvement and adverse effects.[9][10]

Corticostriatothalamocortical (CSTC) circuits passing through subcortical regions are implicated in the pathogenesis of OCD.[11][12] Thus, several subcortical structures, including anterior limb of the internal capsule, ventral capsule/ventral striatum (VC/vs), nucleus accumbens (NAc), bed

nucleus of stria terminalis, anteromedial subthalamic nucleus (amSTN) and inferior thalamic peduncle are targets for DBS in OCD. DBS targeting some of these structures has been found to significantly reduce OCD symptoms compared with sham stimulation.[13] Based on evidence from a double-blinded randomised controlled trial,[14] a recent treatment guideline recommended bilateral STN DBS for treatment-refractory OCD.[15] Two recent randomised crossover trials found equivalent efficacy of DBS targeting amSTN compared with VC/VS, caudate nucleus and NAc.[16 17] In the latter study, most patients, who were masked/blinded to the target of stimulation, preferred the amSTN stimulation based on subjective improvement.[17] A recent systematic review also found similar efficacy for DBS of amSTN and striatal targets.[8] Interestingly, recent evidence suggests that the different targets have similar connections along frontosubcortical circuits and thus may be targeting the same network.[18 19]

The adverse effects of DBS are frequently related to stimulation and generally mild and reversible.[7 20] DBS is an invasive procedure that requires long-term close monitoring for optimisation of stimulation as well as periodic changes in battery, which adds to the treatment cost. Despite these limitations, long-term follow-up studies have shown that DBS leads to improvement in symptoms as well as the quality of life.[21 22] However, DBS is helpful in only 60%–75% of patients with treatment-refractory OCD.[20 22] There is a need to identify predictors of treatment response, which would assist in patient selection for this invasive and expensive treatment.

Despite decades of research, the exact mechanism of action of DBS is still not clearly understood.[23] Recent evidence suggests that DBS has both local and distant effects with resultant amelioration of pathological network activity.[11 23] The role of target nuclei such as amSTN in the pathophysiology of OCD provides useful clues to unravel the mechanism, which would help identify biopredictors of response. Further, DBS provides a unique opportunity to study the role of STN in psychopathology with spatial and temporal precision.

### Neurocircuitry of OCD

Although the aetiology of OCD is still unknown, converging evidence from neuroimaging studies implicates the dysfunctional CSTC circuits in the pathophysiology of OCD.[11 24] Particularly, the CSTC pathways involving the orbitofrontal cortex (OFC) and, to a lesser extent, the dorsal anterior cingulate cortex (dACC) are dysfunctional in OCD.[25–28] Functional neuroimaging studies show that these cortical regions are hyperactive at rest, which is accentuated during symptoms provocation.[29] Pretreatment OFC and caudate metabolism predict response to medications in OCD.[30 31] Similarly, DBS over various targets modulate the activity in these circuits by increasing their connectivity.[32 33] Thus, modulation of functional connectivity in the CSTC circuits is an important target for antiobsessive treatments, including DBS.

### STN in the pathophysiology of OCD

STN is the key basal ganglia input structure of the 'hyperdirect' pathways.[34] Based on hyperdirect connectivity, the STN is divided into three partially overlapping subterritories, namely the limbic, associative and motor regions.[35–37] Computational models and imaging studies suggest that the 'hyperdirect' pathway connecting inferior frontal gyrus (IFG) and STN is involved in global response inhibition, for example, during the stop signal task (SST).[38 39] Functional MRI (fMRI) studies have shown activation of STN, IFG and presupplementary motor area during SST performance.[39 40] STN also plays a crucial role in decision-making and response selection in conflict situations, by setting a decision threshold that is contextually modulated.[41 42] Recent evidence suggests that neurons in ventromedial STN, which is preferentially connected to dACC and OFC, are especially involved in reactive stopping and switching.[36 43] Both these functions are impaired in patients with OCD.[44 45] Following STN DBS, patients with OCD and Parkinson's disease show decreased reaction time and performance, suggesting more 'impulsivity', as defined in response inhibition paradigms.[46] STN DBS modulates uncertainty/conflict-driven decision threshold adjustment and adapting to speed/accuracy trade-offs.[47] Impairment in task switching improves, specifically following stimulation of ventral DBS contacts in STN.[48] Thus, DBS targeting amSTN may decrease decisional threshold towards more optimal levels, leading to a less cautious and more rapid goal-directed behaviour, which may be beneficial in patients with OCD.[37 49]

STN connectivity has been implicated in the pathophysiology of OCD. Resting-state functional connectivity of STN with cortical and striatal structures has been associated with cognitive and behavioural measures of compulsivity.[50] Tracing studies in non-human primates have found hyperdirect pathways connecting OFC, dACC and dorsolateral prefrontal cortex (DLPFC) projecting to the anteromedial STN.[36] A prospectively acquired imaging study found that effective STN DBS targets in OCD are located in this region, that is in the anterior inferior medial border, which has direct connections with the OFC, dACC and DLPFC.[16] Further, evidence suggests that STN DBS modulates corticostriatal connectivity during SST.[49] STN DBS in OCD decreases metabolism in ACC and the therapeutic effects correlate with a decrease in OFC metabolism.[51] Thus, CSTC networks connecting various prefrontal regions with STN play a role in the neurocognitive functions underlying OCD, its behavioural manifestations and may mediate the therapeutic actions of DBS in OCD. The hyperdirect pathways have especially been implicated in this regard.[52]

Another line of evidence has used the unique opportunity provided by DBS to collect electrophysiological data directly from the nucleus. STN oscillatory activity and frontal cortico-STN coherence in β and θ frequency bands are associated with different phases of the SST, with the latter prominently seen in the ventral contacts.[53] The medial PFC-STN θ phase coherence increases during

high-conflict trials in the flanker task.[54] Response inhibition and conflict monitoring, assessed through SST and flanker task, respectively, are putative endophenotypes for OCD.[55 56] In OCD subjects, bursting and oscillatory activity have been localised to the associative/limbic STN, with predominant oscillatory activity in δ-band. Further, the severity of OCD is associated with low frequency oscillatory activity and burst characteristics.[57] Low frequency activity (θ band) in the ventromedial STN has been associated with symptom provocation and cognitive/emotional functioning in OCD subjects.[58] A recent study demonstrated that emotional images distinctly modulated STN θ band activity in OCD subjects and emotion-related θ band activity correlated with symptom severity.[59] Thus, electrophysiological activity in STN, in particular low frequency oscillatory activity in the ventromedial region, is a putative biomarker associated with neurocognitive functions underlying OCD and its clinical manifestations.

Overall, STN-cortical connectivity (especially with the frontal cortex), scalp electroencephalographic activity in the frontal cortex, bursting/oscillatory activity in the STN and STN-related neurocognitive functioning (response inhibition) are potential biomarkers for treatment response to DBS in OCD.

The current proposal plans to identify such biomarkers with the following objectives:

### Primary objective

1. To evaluate whether STN functional connectivity when performing response inhibition task (SST) predicts improvement in obsessive–compulsive symptoms after DBS over STN.

### Secondary objectives

1. To evaluate whether the improvement of obsessive–compulsive symptoms following STN DBS is predicted by
   a. baseline STN resting-state functional connectivity with PFC.
   b. STN functional connectivity with PFC regions during symptom provocation paradigm.
   c. White matter structural connectivity (fractional anisotropy and other measures) of STN with cortical regions.
   d. Local field potential (LFP) time frequency amplitude in the STN at rest, during response inhibition task and symptom provocation.
   e. Frontal electroencephalogram (EEG) activity at rest, during response inhibition task and symptom provocation.
   f. Localisation of DBS electrodes within the STN.
2. To study the neuroadaptive changes following DBS by evaluating neurocognitive performance and EEG activity before and 12 months post DBS.
3. To evaluate the effect of DBS on clinical symptomatology, disability, quality of life and subjective well-being.

We hypothesise that baseline STN prefrontal connectivity, low frequency oscillatory activity over frontal lobes/

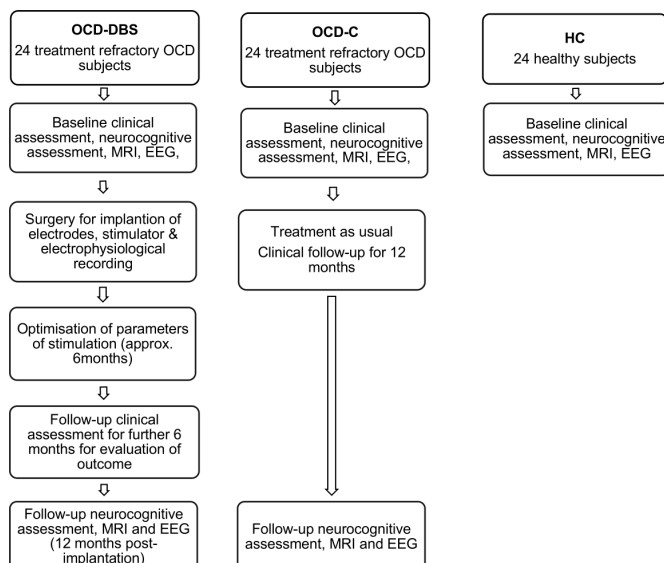

**Figure 1** Schematic representation of study protocol. All subjects would undergo baseline clinical and neurocognitive assessment, MRI and EEG. For the subjects in the OCD-DBS group, that is, OCD subjects who receive DBS, stimulation parameters would be optimised over the next 6 months; followed by another 6 months of clinical follow-up. Subjects in the OCD-C group, that is, subjects who receive treatment as usual, would undergo clinical follow-up for 12 months. Both the OCD groups would undergo neurocognitive assessment, MRI and EEG at the end of 12 months. Subjects in the HC group would not undergo any further assessment following baseline. EEG, electroencephalogram; HC, healthy controls; OCD, obsessive–compulsive disorder; OCD-C, OCD control.

STN and impaired performance on response inhibition task would predict improvement in obsessive–compulsive symptoms after STN DBS in OCD. The current report is prepared based on Standard Protocol Items: Recommendations for Interventional Trials checklist.[60]

### Experimental design

We plan to conduct a longitudinal open-label follow-up study of 24 subjects undergoing STN DBS for severe, chronic and treatment-refractory OCD to evaluate whether baseline biomarkers predict a decrease in the severity of OCD at follow-up. The DBS leads and stimulator would be implanted after baseline investigation for biomarkers. The initial 6 months of follow-up would be used for the optimisation of stimulus parameters. This would be followed by an open-label clinical follow-up of 6 months. The primary outcome would be a clinical improvement after 6 months of stable stimulation, that is, approximately 1 year after the implantation procedure. A comparison group consisting of 24 subjects with chronic, severe and treatment-refractory OCD, who would receive treatment as usual, would also be followed-up for 1 year. We would also recruit a sample of 24 healthy volunteers for comparison of biomarkers at baseline (figure 1). Subject enrolment is expected to commence in July 2021 and complete by September 2024.

## Sample
### DBS group (OCD-DBS)

Twenty-four subjects with severe, chronic and treatment-refractory OCD would be recruited from the OCD clinic, inpatient and outpatient services of the National Institute of Mental Health and Neurosciences (NIMHANS), Bangalore. The subjects would be screened with the following selection criteria:

### Inclusion criteria

1. Age 18–60 years.
2. Primary diagnosis of OCD satisfying the Diagnostic and Statistical Manual of Mental Disorders (DSM)-5 criteria,[61] established using Mini International Neuropsychiatric Interview 7.0.2.[62]
3. Duration of OCD ≥5 years.
4. Yale-Brown Obsessive Compulsive Scale (Y-BOCS)[63] score of ≥28 or ≥14 if the subject has either predominantly obsessions or compulsions alone.
5. Clinical Global Impression-Severity (CGI-S)[64] scale of ≥5.
6. Disability of ≥40% as evaluated by the Indian Disability Evaluation and Assessment Scale.[65]
7. Failure to obtain meaningful improvement in OCD despite adequate trial with standard treatment strategies, which should include:
   a. At least three trials of SRIs, one of which should be clomipramine, which was either ineffective or poorly tolerated. An adequate trial includes the recommended dose of SRIs for ≥12 weeks' duration each.[6]
   b. Augmentation of SRIs with at least two agents for ≥6 weeks, one of which should be either risperidone or aripiprazole.
   c. Trial of structured behaviour therapy/CBT for at least 20 sessions or demonstrated an inability to tolerate the anxiety due to therapy.

### Exclusion criteria

1. Diagnosis of bipolar disorder, a psychotic disorder of ≥3 months' duration as assessed with MINI 7.0.2.
2. Current substance use disorder (except caffeine or nicotine use disorder) or major depressive episode or current high suicidality, as assessed with MINI 7.0.2.
3. Severe personality disorder as assessed with Structured Clinical Assessment of DSM-5-Personality Disorders .
4. Clinically significant abnormality on MRI of the brain.
5. Pregnancy, contraindication for DBS/anaesthesia/preoperative MRI and inability to comply with surgical requirements.

All potential study participants satisfying the above criteria will be assessed for suitability for DBS by two psychiatrists and a neurosurgeon from the study team. An independent review committee with members (consisting of two psychiatrists, one neurologist and one neurosurgeon) who are not a part of the study team would ratify the decision for eligibility for surgery.

### OCD control group

We would recruit 24 OCD subjects fulfilling the above criteria from the same population, including those subjects who refuse consent for DBS. These subjects would receive treatment for OCD as recommended by the treating clinician.

### Healthy volunteers

Twenty-four age and gender-matched healthy volunteers would be recruited from the community to compare the baseline differences in neuroimaging, neurocognitive performance and EEG with the study subjects. They would be screened for the presence of psychiatric disorder using MINI 7.0.2. Other exclusion criteria include Y-BOCS score ≥12, family history of psychiatric disorder in first-degree relatives, clinically significant neurological illness, pregnancy and contraindication for MRI.

### Sample size estimation

For evaluating the abovementioned primary objective, the optimal sample size was estimated using standard principles and methods.[48] Based on approximation and estimation from the previous data on fMRI-based predictors of treatment response for other interventions in OCD (n=12, 15),[49 50] a sample size of at least 24 OCD patients will be required to detect a two-tailed significant difference of $\alpha = 0.05$ with an estimated 90% power for an estimated correlation coefficient of 0.6 (Pearson's) between STN-PFC connectivity and change in Y-BOCS total score with add-on DBS.

### Baseline assessment
#### Clinical assessment

The clinical status would be assessed with the MINI 7.0.2.[62] All OCD subjects would undergo baseline assessment with Y-BOCS, which includes a symptom checklist to assess the nature of symptoms, severity scale to assess the severity of symptoms and insight/avoidance scale to assess these domains of symptoms. The severity of illness would also be assessed with the CGI-S scale. The severity of depressive and anxiety symptoms would be assessed with the Hamilton Depression Rating Scale (HAM-D)[66] and Hamilton Anxiety Rating Scale (HAM-A),[67] respectively. The extent of disability, quality of life and mental well-being would be assessed with the WHO Disability Assessment Schedule 2.0–36 item version (WHODAS 2.0),[68] WHO quality of life instrument (WHOQOL-BREF)[69] and WHO-5 Well-Being Index (WHO-5).[70] The healthy volunteers would undergo screening with MINI 7.0.2 and Y-BOCS.

### Neurocognitive assessment

All the subjects would undergo a detailed neurocognitive evaluation consisting of the modified NIMHANS neuropsychology battery,[71] SST[72] for assessing response inhibition, modified flanker task[55] for assessing response inhibition in a conflict situation/error monitoring, beads task[37 73] for assessing decisional impulsivity and temporal discounting task[74–76] for assessing reward-related decision-making.

---

**Box 1  Neurocognitive and personality assessment**

**Neurocognitive assessment**
1. Modified NIMHANS neuropsychologyneuropsychology battery[71]
   – Finger tapping.
   – Bender-Gestalt test.
   – Digit symbol substitution test.
   – Colour trails 1 and 2.
   – Block design test.
   – Controlled Oral Word Association Test.
   – Animal names test.
   – Digit span.
   – Spatial span.
   – Tower of London.
   – Wisconsin Card Sorting Task.
   – Stroop test.
   – Auditory verbal learning test.
   – Rey's complex figure test.
2. Stop signal task.[72]
3. Modified flanker task.[55]
4. Beads task.[37]
5. Temporal discounting task.[74–76]

**Personality/behavioural assessment**
1. Frontotemporal Behavioural Scale.[78]
2. Iowa Scales for Personality Change.[79]
3. UPPS-P Impulsive Behaviour Scale.[80]

Impairment in the above neurocognitive functions have been observed in OCD patients and they are closely associated with amSTN functioning.[37 43 54–56 77] Besides, Frontotemporal Behavioural Scale,[78] the Iowa Scales for Personality Change[79] and the Urgency-Premeditaton-Perserverance-Sensation seeking-Positive Urgency (UPPS-P) Impulsive Behaviour Scale[80] would be administered to assess for personality/behavioural dysfunction. Subjects from all the three groups would undergo the neurocognitive/behavioural assessment (box 1) at baseline. Besides, OCD subjects would undergo a repeat evaluation at the end of 12 months.

## MRI

MRI will be acquired with a 3T MRI scanner (Philips Ingenia) at NIMHANS, Bangalore. Both groups of OCD subjects and healthy volunteers would undergo a baseline structural MRI of the brain and diffusion tensor imaging (DTI) to evaluate the baseline structural connectivity of the STN. Subjects in the OCD-DBS group would undergo an additional baseline structural MRI with a stereotactic frame attached to guide the accurate targeting of electrodes. All subjects would undergo multiecho (ME) resting-state fMRI of the brain to study the functional connectivity of the STN. The subjects would also undergo a ME-fMRI while performing SST and under symptom provocation (only for the OCD-DBS and OCD control (OCD-C) subjects) employing symptom dimension-specific pictures. ME-fMRI during SST activates the STN and connected cortical regions.[40] Blood oxygenation level-dependent activity in this paradigm would be tested as primary predictors of treatment response to DBS. The symptom provocation paradigm activates corticostriatal and limbic structures.[81] Activity in these regions has been found to predict response to medications and CBT.[82 83] To the best of our knowledge, ME-fMRI has not been performed under symptom provocation. Hence, it is unclear whether this paradigm would activate STN. Due to the previous data showing activation of OCD-specific circuits, activity under this paradigm will be tested as an additional predictor of treatment response.

## Electroencephalography

All subjects would undergo an awake 64-channel EEG as per the international 10–20 EEG system electrode placement, in a comfortable resting supine position. Recordings from the frontocentral (FCz), central (Cz) and bilateral dorsolateral prefrontal (F3 and F4) montages would be the focus of analysis. The OCD subjects and healthy volunteers would undergo EEG while performing the SST and flanker tasks. The OCD subjects would also undergo EEG symptom provocation using symptom dimension-specific pictures. Artifact-free epochs of resting-state EEG epochs will be selected for power spectral analysis, which involves examining the spectral power in various frequency bands ($\delta$=1–3 Hz, $\theta$=4–7 Hz, $\alpha$=8–11 Hz, $\beta$=12–30 Hz and $\gamma$>30 Hz) at each electrode. For examining the EEG under symptom provocation, power spectral analysis would be conducted on 2 s epochs selected from 1 s before stimuli to 1 s after the stimuli that is, symptom dimension-specific pictures.

## Deep brain stimulation

Following the baseline assessments, subjects in the OCD-DBS will undergo bilateral STN DBS.

### Surgical procedure

STN would be located through direct visualisation in MR images. Quadripolar electrodes would be implanted at the boundary of limbic and associative territories of STN, bilaterally, under local anaesthesia and antiseptic precautions. The amSTN would be targeted 2 mm anterior and 1 mm medial to the motor STN target used for DBS in Parkinson's disease.[14] The targets would be refined through intraoperative microrecording along multiple trajectories.[84] Intraoperative macrostimulation would be conducted to evaluate the acute effects of stimulation. The trajectory would be chosen based on acute adverse vis-a-vis beneficial effects. The position of the electrodes would be confirmed by postoperative MRI using a 3D atlas MRI co-registration.[85] Electrophysiological activity from STN would be recorded perioperatively and postoperatively. The stimulator would be implanted subcutaneously 5–7 days following STN stimulation after electrophysiological recording.

### Electrophysiological recording from STN

Perioperative single-unit recording would be recorded through microelectrodes used for STN localisation. Spiking activity would be recorded during 2 min of

rest and symptom provocation through stereotactically lowered microelectrode leads from various regions of the STN. Symptom provocation would be performed by showing individualised symptom-provoking pictures. LFPs would be recorded through the DBS stimulating electrodes postoperatively. LFPs would be recorded through bipolar montages (to reduce volume conductance effect) from all adjacent contacts, yielding three channels/hemisphere. The electrodes would be externalised through the scalp with extension cables connected to an amplifier for LFP recording. Adequate precautions would be taken to prevent postoperative infection. The LFP recording would be performed on postoperative days 3–5 at rest, during SST, flanker task and under symptom provocation. Power spectral analysis would be performed to study oscillatory activity in various frequency bands ($\delta$=1–3 Hz, $\theta$=4–7 Hz, $\alpha$=8–11 Hz, $\beta$=12–30 Hz and $\gamma$>30 Hz).

### Optimisation of stimulation parameters

After the implantation of the stimulator, programming would be finalised over the subsequent 6–8 months. The stimulator would be set at a frequency of 130 Hz and pulse width of 60 µs. Successive trials of monopolar stimulation from each contact would be attempted, beginning with the most ventral contact. Voltage would be increased gradually to monitor for adverse effects. Each contact would be evaluated for 6–8 weeks to evaluate therapeutic efficacy. The final contact would be chosen based on a balance between clinical efficacy and adverse effects.

### Follow-up intervention

Following the optimisation of parameters that may take up to 6 months postimplantation of stimulator and electrodes, so the patient would be followed up for another 6 months, considering the latency in onset of improvement with DBS in OCD. DBS would be provided as an unblinded add-on treatment. Although the subjects chosen are treatment-refractory, medication changes and CBT would be permitted during the follow-up period due to ethical concerns as well as to improve the external validity of the study. CBT would also be permitted during follow-up as it may improve the outcome after DBS.[86]

Subjects in the OCD-C would undergo the best alternative treatment for treatment-refractory OCD based on the standard treatment practices of OCD clinic, NIMHANS, and evidence-based clinical practice guidelines. The participants would be unblinded to the group allotment. If they wish to undergo neurosurgical interventions, including DBS for OCD during follow-up, they would be offered the same and would be removed from the control arm of the study.

### Follow-up assessments
#### Clinical assessments

Subjects from both OCD-DBS and OCD-C groups would undergo structured clinical assessment during follow-up. Assessment during follow-up would include a clinical evaluation consisting of monthly assessment of Y-BOCS,

CGI-S, HAM-D and HAM-A (box 1). The subjects would do a self-assessment of their obsessive–compulsive symptoms using a smartphone-based app. WHODAS 2.0, WHOQOL-BREF and WHO-5 would be administered every 3 months. Y-BOCS scores at the end of 6 months of follow-up postoptimisation of stimulation parameters would be the primary outcome variable. The structured assessment would be performed through an in-person interview or video conferencing. Subjects in the OCD-C group would also be assessed with same baseline and follow-up clinical assessments, as specified above.

### Neurocognitive assessment, MRI and EEG

At the end of 6 months of stable stimulation (ie, approximately 12 months postimplantation), both groups of OCD subjects would undergo a detailed neurocognitive assessment as well as EEG (using the same paradigms described above). Subjects in the OCD-DBS group would undergo follow-up EEG both in the stimulation turned 'ON' and 'OFF' phases on 2 consecutive days. The DBS artifacts in the follow-up EEG recordings during the 'ON' phase would be removed using recommended techniques, including oversampling, temporal low-pass filtering and frequency-domain Hampel filtering.[87] The effect of DBS on neurocognitive and EEG markers would be evaluated and correlated with symptom change. Acquiring MRI scans with DBS electrodes in situ may carry the risk of image distortion, interference with DBS stimulator and risk of tissue damage due to overheating of the electrodes.[88] However, recent reports suggest that MRI scans could be obtained safely under appropriate precautions.[89–91] DBS manufacturers recommend precautions for acquiring MRI scans in this population, such as acquisition using 1.5T MRI. Thus, postsurgical MRI scans (including T1-weighted structural images, fMRI and DTI) would be acquired under appropriate precautions, only for subjects who provide informed consent after explaining the risks and benefits associated with the procedure. Subjects in the OCD-C group would undergo follow-up MRI, EEG and neurological assessments at the end of 12 months, using the same paradigms and protocols as in the OCD-DBS group. As the subjects would not undergo surgery, the electrophysiological recording would not be obtained from the STN. The assessment schedule for the three groups is shown in table 1.

The OCD-C subjects would serve as a comparator for the intervention group to establish the efficacy of DBS. The baseline and follow-up biomarkers would be compared between the two groups using the standard statistical techniques mentioned in the protocol. The healthy volunteer group would serve as a comparator to establish the baseline differences in MRI, EEG and neurocognitive test performances with the OCD subjects.

The statistical analysis would be conducted using the SPSS software (SPSS, Chicago, Illinois, USA). The fMRI processing and analysis would be performed using the Statistical Parametric Mapping (SPM) software (http://www.fil.ion.ucl.ac.uk/spm/). Functional connectivity

**Table 1** Assessment schedule

| | OCD intervention group | OCD control group | Healthy volunteers |
|---|---|---|---|
| MINI 7.0.2 | Baseline | Baseline | Baseline |
| Y-BOCS | Baseline+every month | Baseline+every month | Baseline |
| CGI-S | Baseline+every month | Baseline+every month | Baseline |
| HAM-A | Baseline+every month | Baseline+every month | Baseline |
| HAM-D | Baseline+every month | Baseline+every month | Baseline |
| WHOQOL-BREF | Baseline+every 3 months | Baseline+every 3 months | Baseline |
| WHODAS 2.0 | Baseline+every 3 months | Baseline+every 3 months | Baseline |
| WHO-5 | Baseline+every 3 months | Baseline+every 3 months | Baseline |
| Neurocognitive testing | Baseline+end of 12 months | Baseline+end of 12 months | Baseline |
| MRI | Baseline+end of 12 months* | Baseline+end of 12 months | Baseline |
| EEG | Baseline+end of 12 months | Baseline+end of 12 months | Baseline |
| Subthalamic nucleus recording | Baseline | x | x |

* Would be acquired after obtaining separate informed consent
CGI-S, Clinical Global Impressions-Severity;[64] EEG, electroencephalography; HAM-A, Hamilton Anxiety Rating Scale;[67] HAM-D, Hamilton Depression Rating Scale;[66] MINI 7.0.2[62], Mini International Neuropsychiatric iInterview; OCD, obsessive–compulsive disorder; WHO-5, WHO-5Five Well-Being Index;[70] WHODAS 2.0, WHO Disability Assessment Schedule;[68]WHOQOL-BREF, WHO quality of life instrument;[69] Y-BOCS, Yale -Brown Obsessive Compulsive Scale.[63]

analysis would be conducted using SPM toolboxes and FMRIB Software Library (FSL) (https://fsl.fmrib.ox.ac.uk/fsl/fslwiki/). Resting-state fMRI would be analysed using SPM, FSL and Data Processing & Analysis for Brain Imaging (DPABI) toolbox (http://rfmri.org/dpabi) according to established processing pipelines. DBS electrodes would be localised from postoperative and preoperative images following standardised pipelines using appropriate toolboxes (eg, lead DBS).[92] The structural connectivity of volume of tissue activated and location of active electrode within STN would be evaluated as a potential predictor of outcome. EEG analysis and electrophysiological analysis would be conducted using appropriate software. The demographic, clinical (including the principal symptom dimension and age-at-onset), neurocognitive and electrophysiological data will be examined for normality. Correlational followed by multiple regression analysis would be used to assess whether the biomarkers predict clinical response (Y-BOCS score) at the end of 6 months of stable stimulation. Similar analyses would be conducted to test the other study objectives.

### Ethical considerations
The study proposal and the informed consent forms have been approved by NIMHANS Ethics Committee (No.NIMHANS/EC(BEH.SC.DIV; 19th meeting dated 16 July 2019). The trial has been registered in the Clinical Trial Registry of India (CTRI/2020/03/024131)-pre-results stage. The subjects would be recruited after obtaining written informed consent explaining the risks and benefits of participating in the study. Subjects in the OCD-DBS group would be diligently educated about and monitored for adverse effects by trained mental health professionals. If subjects from the OCD-C group wish to undergo neurosurgical interventions for OCD during follow-up, they would be offered the same and would be removed from the control arm. Postoperative MRI would be conducted only in consenting subjects from the OCD (intervention) group after explaining the risks and benefits. The subjects in the intervention group would be protected by an insurance plan funded by the research proposal during the process of the study. A data safety monitoring board would be constituted to periodically review the progress of the study to assess safety issues and recommend continuation, modification and termination of the trial as required. All OCD subjects can continue to receive treatment from the OCD clinic, NIMHANS, following the study proposal irrespective of whether they complete the study.

### Dissemination
The data will be anonymised, curated and stored in a computer hard disk, with appropriate backup. In keeping with the policies of the funding agency, the data and metadata would be shared with the wider research community through appropriate platforms. The study findings would be disseminated in peer-reviewed scientific publications and scientific conferences, irrespective of outcome.

### Expected outcome
Identification of biomarkers of treatment response to bilateral STN DBS in OCD would pave way for personalised medicine by assisting patient selection for this invasive and expensive treatment. Furthermore, this would help understand the mechanisms of action of treatment, which would assist in future refinement/optimisation of treatment and open the scope for innovations, including closed-loop stimulation.

## Author affiliations

[1]Department of Psychiatry, National Institute of Mental Health and Neurosciences, Bangalore, Karnataka, India
[2]Department of Neurosurgery, National Institute of Mental Health and Neurosciences, Bangalore, Karnataka, India
[3]Department of Clinical Psychology, National Institute of Mental Health and Neurosciences, Bangalore, Karnataka, India
[4]Univ Paris-Est Créteil, DMU CARE - Département Médical-Universitaire de Chirurgie et Anesthésie réanimation, DMU IMPACT, Département Médical-Universitaire de Psychiatrie et d'Addictologie, Hôpitaux Universitaires Henri Mondor, Creteil, France
[5]Univ of Paris 12 UPEC, Faculté de médecine, INSERM U955, Creteil, France
[6]Institut du Cerveau, ICM, Inserm U 1127, CNRS UMR 7225, Sorbonne Université, Paris, France
[7]Department of Mental Health and Psychiatry, University of Geneva, Geneva, Switzerland

**Acknowledgements** GV acknowledges the support of the DBT Wellcome Trust India Alliance Clinical/Public Health Research Centre Grant (IA/CRC/19/1/610005).

**Contributors** SSA, the recipient of the Indian Alliance Fellowship, conceived and drafted the protocol. YCJR advised on study design, clinical phenotyping, funding application and mentored the primary applicant. GV mentored funding application and mentored the primary applicant, advised on study design, neuroimaging and electrophysiological recording. LM mentored the applicant, is the external sponsor for the fellowship, advised on study design, finalising biomarkers and deep brain stimulation (DBS) targeting/procedure. PD advised on study design, DBS management, electrophysiological recording, biomarkers and DBS targeting/procedure. DS is a collaborator in the project advised on DBS procedures and study design. JCN is a collaborator who worked on the study design, advised on neuroimaging, electroencephalogram, neurocognitive assessment and study design. TSJ advised on clinical phenotyping, study design and clinical recruitment. SP advised on DBS targeting and neurosurgical procedures. HK advised on neurocognitive assessment. All authors contributed to the refinement of the study protocol and approved the final version.

**Funding** The study is fully funded by the DBT/Wellcome Trust India Alliance Intermediate fellowship granted to SSA. All the material/equipment, including deep brain stimulation electrodes, would be purchased through funds from the fellowship.

**Competing interests** None declared.

**Patient and public involvement** Patients and/or the public were not involved in the design, or conduct, or reporting, or dissemination plans of this research.

**Patient consent for publication** Not required.

**Provenance and peer review** Not commissioned; externally peer-reviewed.

**ORCID iDs**
Shyam Sundar Arumugham http://orcid.org/0000-0002-5641-454X
TS Jaisoorya http://orcid.org/0000-0003-0322-9209

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
