## [Reviewer comments · BMJ Open]

ARTICLE DETAILS

TITLE (PROVISIONAL)	Identification of biomarkers that predict response to subthalamic nucleus deep brain stimulation in resistant obsessive-compulsive disorder: protocol for an open-label follow-up study
AUTHORS	Arumugham, Shyam Sundar; Srinivas, Dwarakanath; Narayanaswamy, Janardhanan; Jaisoorya, TS; Kashyap, Himani; Domenech, Philippe; Palfi, Stéphane; Mallet, Luc; Venkatasubramanian, Ganesan; Reddy, YC Janardhan

VERSION 1 – REVIEW

REVIEWER	Herz , Damian University of Oxford
REVIEW RETURNED	21-Jan-2021

GENERAL COMMENTS	In this study protocol Arumugham and colleagues wish to study putative ‘biomarkers’ that predict the clinical outcome of deep brain stimulation (DBS) of the subthalamic nucleus (STN) in patients with obsessive-compulsive disorder (OCD). DBS is a promising treatment option in OCD and a further understanding of its behavioral and neurophysiological effects as well as putative predictors of responders vs. non-responders would be very useful. The rationale for the study is well motivated, the background, objectives and design are clearly described. The study design (open-label follow-up study with a non-DBS OCD as well as healthy control group) seems appropriate. I have some comments, which might help to improve the protocol and conductance of the study. (i) Currently the hypotheses about predicting factors for DBS outcomes all relate to different patient ‘phenotypes’. I would suggest including localization of the STN electrodes as well, since a suboptimal placement might lead to an unsuccessful outcome even if the patient ‘phenotype’ might be well suited for DBS. (ii) The description of the EEG implies that only 4 electrodes will be used (Fz, Cz, F3, F4). If this indeed is intended, I would suggest a broader coverage with higher density EEG systems in order to increase the signal to noise ratio, even if the main focus of the analysis would be on the frontal electrodes. (iii) Will the follow-up EEG be conducted during DBS? If so, please detail how to clean the data from the DBS-artefacts. This might also make the follow-up EEG comparison to the non-DBS group difficult.
---

	(iv) Is it correct that the fellowship of the first author will be used to fund all costs related to the project including the DBS electrodes (see page 4 'Funding')? Would the clinical hardware not be paid for through health insurances / the health care system? If the DBS hardware indeed is only funded through the fellowship grant, how will the long-term management of the patients be secured, e.g. replacement of the IPG? (v) Please clarify in the 'Dissemination' segment that the results will be published irrespective of the outcome, i.e. that both positive and negative findings will be published.
--	---

REVIEWER	Turner, Dennis Duke University Department of Neurobiology, Neurosurgery, Biomed Engineering
REVIEW RETURNED	30-Jan-2021

GENERAL COMMENTS	This manuscript is a planned open-label trial study protocol of anterior medial STN DBS (amSTN DBS) for OCD. OCD patients will be selected for possible entry, then if disinterested can participate in a companion cohort who do not undergo prospective assessment and analysis of "biomarkers". These will include tract-tracing MRI and neurocognitive assessment whereas both intraoperative microelectrode recording and postoperative local field potential recordings will be performed from percutaneous extensions on the DBS electrodes. An additional cohort of 24 control subjects will also be recruited for the non-invasive biomarkers. This is an innovative study approach with 2 control populations to the open label DBS. I would disagree with the Abstract opening statement: DBS within ant medial STN may be effective in small trials – "proven to be effective" is a strong statement, transcending available evidence. The introduction is excellent with a good summary, but data remains unclear from all reviews and studies as to whether ant medial STN is really any better than anterior VC (in the combined study using both sites improved outcome further). Since there has not been a larger, confirming randomized trial (ie, > 6-8 patients) the relative efficacy of the various sites remains an open question. Most of the references are up to date and relevant but a more recent review on prevalence and treatment options may be preferred: Stein DJ et al, Nature Reviews 5: 52, 2019 vs 2005 article from Brazil. Specific subtypes of OCD are not characterized in this proposed trial (ie, hoarding, checking, body dysmorphic, symmetry, contamination, etc) whereas this remains controversial in the DBS outcome (ie, hoarders do not respond as well as other groups). This would be extremely helpful for the field, since this would be the largest prospective trial of OCD to date. The nomenclature (to an international audience reading English) is slightly confusing: the authors use the word DBS "over" STN whereas it would make much more sense to be specific: DBS "within" anterior medial STN borders. The sample size estimation is confusing: the general response rate of any site in OCD is ~ 45% (from clinical evaluation) whereas this trial is based on fMRI predictors of treatment response. What would be the sample size if using prelim data estimated from the prior randomized trial or other prospective trials? This is non-traditional, using an fMRI marker rather than a clinical domain, for primary outcome. Many of the impulsivity tasks and response inhibition tasks are more specific to dopamine agonist treatments in Parkinson's
---

	disease rather than DBS, even in the motor or cognitive regions of STN. Are these tasks commonly used in OCD and are they specific to anterior medial STN? In the paragraph describing these studies (page 7 lines 13-43) there is considerable confusion amongst different regions of STN and their resulting effects – more specificity would be welcomed. In the experimental design (p 8, line 47-48) “open-label naturalistic clinical follow-up” is described – what does this mean? Is stimulation still on or turned off (ie, alluding to “natural history”)? With the EEG will the stimulation be “on” or “off” – how will interference be overcome? Specific starting coordinates (wrt AC-PC coordinates and ordinary motor STN) should be specified. Though there is indication that the targets would be “defined” through microelectrode recordings there is no description as to how this will be achieved. Typically recordings “refine” an initial target with specific plans for how to improve the target. How will “symptom provocation” be performed in the OR? The percutaneous recordings will be very helpful but further description as to how they are performed should be added – are they differential, adjacent contacts or single contacts wrt a distant ground? Will all contacts be tried? Since this is a trial, there should be some guidance as to how stim parameters will be honed for optimal clinical response – is there a protocol? How long between adjusting parameters? This topic has plagued DBS adjustment for OCD since may be difficult to know how well parameters work for a delay. Ethical considerations will allow cross over (ie, p 13). Why not have them wait the proscribed year of f/u before cross over allowed to stimulation (since asymmetrical – will cross overs to no stim be allowed)? This is standard procedure in most experimental studies. Overall, very interesting. Would provide much higher data if randomized and there is no clear reason why this could not be accomplished within this setting. Likewise, in many locations VC is still the preferred DBS target – why not a superset comparing the two, where combined stimulation may be better? It would be extremely helpful to derive some specificity from the hyperdirect frontal lobe targets of STN, differentiating OCD from depression and other diagnoses which also appear to be controlled from dorsolateral prefrontal cortex – this could be a secondary outcome if possible. The limits of the hyperdirect projections from frontal lobe to STN are still being established, perhaps a measure of density of projections from various regions.
--	--

VERSION 1 – AUTHOR RESPONSE

Reviewer 1:

1. Currently the hypotheses about predicting factors for DBS outcomes all relate to different patient ‘phenotypes’. I would suggest including localization of the STN electrodes as well, since a suboptimal placement might lead to an unsuccessful outcome even if the patient ‘phenotype’ might be well suited for DBS

Response: We agree that electrode localisation is an important predictor and have included it as a secondary objective (page-8, point f. under “secondary objective”). We have also elaborated on this

aspect in page 11, “Plan for analysis” – second paragraph.

2. The description of the EEG implies that only 4 electrodes will be used (Fz, Cz, F3, F4). If this indeed is intended, I would suggest a broader coverage with higher density EEG systems in order to increase the signal to noise ratio, even if the main focus of the analysis would be on the frontal electrodes.

Response: We had initially planned post-operative recording of EEG in the OCD-DBS group to allow simultaneous recording of EEG and LFP. Hence, we planned recording from limited EEG electrodes due to limitations posed by surgical wound and dressing. On further deliberation, we decided to record EEG pre-operatively to minimise the risk of postoperative infection. Based on the current timing of recording and reviewer’s suggestion, we would be able to perform a higher density EEG with a 64-channel EEG. We have modified the description in page 9, under the subheading “Electroencephalography”

3. Will the follow-up EEG be conducted during DBS? If so, please detail how to clean the data from the DBS-artefacts. This might also make the follow-up EEG comparison to the non-DBS group difficult.

Response: The follow-up EEG would be conducted in the OCD-DBS group, with the stimulation turned “ON” and “OFF” on 2 consecutive days. The DBS-artefacts in the follow-up EEG recordings during the stimulation “ON” phase would be removed using recommended techniques including oversampling, temporal low-pass filtering, frequency domain hampel filtering etc. .[1] We understand that the solution may not be perfect and hence the results would be interpreted cautiously. These aspects have been discussed in page 10 under “Follow-up assessments”, subheading “Neurocognitive assessment, MRI and EEG”

4. Is it correct that the fellowship of the first author will be used to fund all costs related to the project including the DBS electrodes (see page 4 ‘Funding’)? Would the clinical hardware not be paid for through health insurances / the health care system? If the DBS hardware indeed is only funded through the fellowship grant, how will the long-term management of the patients be secured, e.g. replacement of the IPG?

Response: The DBS electrodes and batteries would be funded through the fellowship grant. At present, there is limited scope for covering the hardware cost through health insurance or public health care system in India. The long term costs such as replacement of IPG has to be borne by the patient or health care insurance, if feasible. These aspects would be discussed with the subjects before enrolment as a part of the informed consent process.

5. Please clarify in the ‘Dissemination’ segment that the results will be published irrespective of the outcome, i.e. that both positive and negative findings will be published.

Response: This has been added in page 11 under the subheading “Dissemination”.

Reviewer 2:

1. I would disagree with the Abstract opening statement: DBS within ant medial STN may be effective in small trials – “proven to be effective” is a strong statement, transcending available evidence.

Response: We have modified the sentence to read as “has been found to be helpful in a subset of patients with...”

2. The introduction is excellent with a good summary, but data remains unclear from all reviews and studies as to whether ant medial STN is really any better than anterior VC (in the combined study using both sites improved outcome further). Since there has not been a larger, confirming randomized trial (ie, > 6-8 patients) the relative efficacy of the various sites remains an open question.

Response: We agree that current evidence does not favour one target over the other. In the introduction, we have cited two randomized crossover trials and a systematic review to emphasize the lack of evidence for differential efficacy between the targets (introduction second paragraph). We have chosen the target in the current study based on recommendation from a recent evidence-based guideline [2] and possibly longer battery life associated with amSTN DBS.[3]

3. Most of the references are up to date and relevant but a more recent review on prevalence and treatment options may be preferred: Stein DJ et al, Nature Reviews 5: 52, 2019 vs 2005 article from Brazil.

Response: We have modified the reference and cited the recent Stein et al article. Introduction line 1.

4. Specific subtypes of OCD are not characterized in this proposed trial (ie, hoarding, checking, body dysmorphic, symmetry, contamination, etc) whereas this remains controversial in the DBS outcome (ie, hoarders do not respond as well as other groups). This would be extremely helpful for the field, since this would be the largest prospective trial of OCD to date.

Response: Thank you for the suggestion. The principal symptom dimension would be evaluated using the Y-BOCS symptom checklist. The principal symptom dimension along with other important clinical variables (such as age at onset, comorbidities etc.) would also be evaluated as potential predictors of outcome. Page 11, subheading "plan for analysis"

5. The nomenclature (to an international audience reading English) is slightly confusing: the authors use the word DBS "over" STN whereas it would make much more sense to be specific: DBS "within" anterior medial STN borders.

Response: We have modified the terminology throughout the manuscript, including title and abstract.

6. The sample size estimation is confusing: the general response rate of any site in OCD is ~ 45% (from clinical evaluation) whereas this trial is based on fMRI predictors of treatment response. What would be the sample size if using prelim data estimated from the prior randomized trial or other prospective trials? This is non-traditional, using an fMRI marker rather than a clinical domain, for primary outcome.

Response: The primary objective of the study is to identify biomarkers rather than clinical response. Hence, we calculated the sample size based on the estimated correlation between degree of functional connectivity and change in Y-BOCS score (both continuous variables).

7. Many of the impulsivity tasks and response inhibition tasks are more specific to dopamine agonist treatments in Parkinson's disease rather than DBS, even in the motor or cognitive regions of STN. Are these tasks commonly used in OCD and are they specific to anterior medial STN?

Response: We chose the stop signal task (SST) and other cognitive tasks based on the following evidence:

Response inhibition is a commonly evaluated neurocognitive marker in OCD, with impairment OCD patients showing impairment in the stop signal reaction time.[4,5] A recent meta-analysis comparing

executive functions in OCD subjects vs healthy controls found the largest effect size for stop signal reaction time.[6] Functional imaging correlates of SST are potential endophenotypes in OCD.[7,8] Multiple lines of evidence implicate STN in SST performance. [9] In particular, neurons in amSTN have been implicated in reactive stopping.[10] Patients with OCD and Parkinson's disease show decreased reaction time in the SST following STN DBS.[11,12] Similarly, flanker task has been employed to evaluate error monitoring in OCD subjects and following STN DBS. [13,14] Beads task and temporal discounting task are also commonly employed in OCD studies to study decisional impulsivity[15,16]. Connectivity of amSTN correlates with beads task performance and the performance is modulated by amSTN DBS in OCD subjects.[16]

These aspects have been discussed in the introduction, page 5 under the subheading "STN in the pathophysiology of OCD". Further, we have added the rationale for choosing the tasks along with the relevant references in the methods in page 8, under the subheading "neurocognitive assessment".

8. In the paragraph describing these studies (page 7 lines 13-43) there is considerable confusion amongst different regions of STN and their resulting effects – more specificity would be welcomed.

Response: We have modified the paragraph to highlight the role of different subregions of STN in OCD relevant cognitive functioning. (Introduction under the subheading "STN in the pathophysiology of OCD)

9. In the experimental design (p 8, line 47-48) "open-label naturalistic clinical follow-up" is described – what does this mean? Is stimulation still on or turned off (ie, alluding to "natural history")?

Response: The stimulation would be turned "ON" during the follow-up phase. By naturalistic follow-up, we meant that medication changes and psychological interventions would be permitted during follow-up in both groups at the discretion of treating clinician and patient. We have removed the word "naturalistic" to avoid confusion.

10. With the EEG will the stimulation be "on" or "off" – how will interference be overcome?

Response: The follow-up EEG would be conducted in the OCD-DBS group, with the stimulation "ON" and "OFF" on 2 consecutive days. The DBS-artefacts in the follow-up EEG recordings during the "ON" phase would be removed through temporal low pass filtering and other recommended techniques.[1] We understand that the solution may not be perfect and hence the results would be interpreted cautiously. These aspects have been discussed in page 10 under "Follow-up assessments", subheading "Neurocognitive assessment, MRI and EEG"

11. Specific starting coordinates (wrt AC-PC coordinates and ordinary motor STN) should be specified. Though there is indication that the targets would be "defined" through microelectrode recordings there is no description as to how this will be achieved. Typically recordings "refine" an initial target with specific plans for how to improve the target.

Response: The STN would be located through direct visualisation in MRI. The amSTN would be targeted 2 mm anterior and 1 mm medial to the motor STN target used for DBS in Parkinson's disease. The targets would "refined" through intraoperative microelectrode recording. We have modified the description of targeting in the manuscript (page 9, under subheading "surgical procedure").

12. How will "symptom provocation" be performed in the OR?

Response: Symptom provocation would be done in the OR by showing the subjects individualized

symptom provocation pictures. Page 9, subheading: electrophysiological recording from STN.

13. The percutaneous recordings will be very helpful but further description as to how they are performed should be added – are they differential, adjacent contacts or single contacts wrt a distant ground? Will all contacts be tried?

Response: The local field potentials would be recorded through bipolar montages (to reduce volume conductance effect) from all adjacent contacts, yielding 3 channels per hemisphere. This has been described in page 9, subheading: electrophysiological recording from STN.

14. Since this is a trial, there should be some guidance as to how stim parameters will be honed for optimal clinical response – is there a protocol? How long between adjusting parameters? This topic has plagued DBS adjustment for OCD since may be difficult to know how well parameters work for a delay.

Response: The stimulator would be set at a frequency of 130 Hz and pulse width of 60 μ s. Successive trials of monopolar stimulation from each contact would be attempted, beginning with the most ventral contact. Voltage would be increased gradually to monitor for adverse effects. Each contact would be evaluated for 6-8 weeks to evaluate therapeutic efficacy. The final contact would be chosen based on a balance between clinical efficacy and adverse effects. We have modified the description - page 10 under the subheading “optimisation of stimulation parameters”.

15. Ethical considerations will allow cross over (ie, p 13). Why not have them wait the proscribed year of f/u before cross over allowed to stimulation (since asymmetrical – will cross overs to no stim be allowed)? This is standard procedure in most experimental studies.

Response: Considering the long follow-up period (1 year) and paucity of alternate interventions, we wanted to keep the option flexible for the OCD-C group. Subjects with severe and treatment refractory OCD may not be willing to wait for 1-year with no stimulation before crossing over to the OCD-DBS group. Similarly, subjects in the OCD-DBS group may not be willing to go back to long-term open-labelled “OFF” stimulation follow-up. Hence we have not considered crossover study.

16. Would provide much higher data if randomized and there is no clear reason why this could not be accomplished within this setting.

Response: We understand that a randomized blinded trial would be methodologically rigorous. However, considering the long-term follow-up period, an open-labelled study would improve recruitment and decrease attrition. It would be difficult to retain subjects over such long periods in a randomized or crossover trial in a refractory population. Further, a change in study design is not feasible at this juncture, as the protocol has been approved for funding.

17. Likewise, in many locations VC is still the preferred DBS target – why not a superset comparing the two, where combined stimulation may be better?

Response: We agree that comparing the VC/VS and STN targets is an important research question. Evaluating this aspect is beyond the scope of the study and would decrease the sample size for evaluating predictors of STN stimulation. The biomarkers evaluated in the current study were chosen based on their relation to STN functioning. Combined stimulation, if employed would require implantation of multiple DBS electrodes in each patient and a recent study found no additional benefit with combined stimulation.[17] Further, major modification of the protocol may be difficult at this stage for reasons mentioned earlier.

18. It would be extremely helpful to derive some specificity from the hyperdirect frontal lobe targets of STN, differentiating OCD from depression and other diagnoses which also appear to be controlled from dorsolateral prefrontal cortex – this could be a secondary outcome if possible. The limits of the hyperdirect projections from frontal lobe to STN are still being established, perhaps a measure of density of projections from various regions.

Response: A recent study evaluating connectivity analysis across four different OCD-DBS cohorts found that the hyperdirect pathway connecting the dorsal anterior cingulate cortex (dACC) to STN may be a common pathway underlying different DBS targets for OCD [18]. As pointed by the reviewer, the specificity of such pathways to OCD has not been established. This may be beyond the scope of the current study, where we recruit only OCD subjects. To obtain more specific results towards OCD symptom reduction, the analysis of association between Y-BOCS symptom reduction and MRI based connectivity measures would be statistically controlled for depression severity scores. Although a correlation of connectivity measures with Y-BOCS and HAM-D would be possible, it would be limited by the exclusive OCD diagnosis in our sample.

References:

- 1 Lio G, Thobois S, Ballanger B, et al. Removing deep brain stimulation artifacts from the electroencephalogram: Issues, recommendations and an open-source toolbox. *Clinical Neurophysiology* 2018;129:2170–85. doi:10.1016/j.clinph.2018.07.023
- 2 Staudt MD, Pouratian N, Miller JP, et al. Congress of Neurological Surgeons Systematic Review and Evidence-Based Guidelines for Deep Brain Stimulations for Obsessive-Compulsive Disorder: Update of the 2014 Guidelines. *Neurosurgery* 2021;88:710–2. doi:10.1093/neuros/nyaa596
- 3 Mallet L, Montcel STD, Clair A-H, et al. Long-term effects of subthalamic stimulation in Obsessive-Compulsive Disorder: Follow-up of a randomized controlled trial. *Brain Stimulation: Basic, Translational, and Clinical Research in Neuromodulation* 2019;12:1080–2. doi:10.1016/j.brs.2019.04.004
- 4 Chamberlain SR, Fineberg NA, Blackwell AD, et al. Motor Inhibition and Cognitive Flexibility in Obsessive-Compulsive Disorder and Trichotillomania. *AJP* 2006;163:1282–4. doi:10.1176/ajp.2006.163.7.1282
- 5 Robbins TW, Vaghi MM, Banca P. Obsessive-Compulsive Disorder: Puzzles and Prospects. *Neuron* 2019;102:27–47. doi:10.1016/j.neuron.2019.01.046
- 6 Snyder HR, Kaiser RH, Warren SL, et al. Obsessive-compulsive disorder is associated with broad impairments in executive function: A meta-analysis. *Clin Psychol Sci* 2015;3:301–30. doi:10.1177/2167702614534210
- 7 de Wit SJ, de Vries FE, van der Werf YD, et al. Presupplementary motor area hyperactivity during response inhibition: a candidate endophenotype of obsessive-compulsive disorder. *Am J Psychiatry* 2012;169:1100–8. doi:10.1176/appi.ajp.2012.12010073
- 8 van Velzen LS, de Wit SJ, Ćurčić-Blake B, et al. Altered inhibition-related frontolimbic connectivity in obsessive-compulsive disorder. *Hum Brain Mapp* 2015;36:4064–75. doi:10.1002/hbm.22898
- 9 Bonnevie T, Zaghoul KA. The Subthalamic Nucleus: Unravelling New Roles and Mechanisms in the Control of Action. *Neuroscientist* 2018;:1073858418763594. doi:10.1177/1073858418763594
- 10 Pasquereau B, Turner RS. A selective role for ventromedial subthalamic nucleus in inhibitory control. *eLife Sciences* 2017;6:e31627. doi:10.7554/eLife.31627
- 11 Obeso I, Wilkinson L, Rodríguez-Oroz M-C, et al. Bilateral stimulation of the subthalamic nucleus has differential effects on reactive and proactive inhibition and conflict-induced slowing in Parkinson's disease. *Exp Brain Res* 2013;226:451–62. doi:10.1007/s00221-013-3457-9
- 12 Kibleur A, Gras-Combe G, Benis D, et al. Modulation of motor inhibition by subthalamic stimulation in obsessive-compulsive disorder. *Transl Psychiatry* 2016;6:e922. doi:10.1038/tp.2016.192
- 13 Zavala B, Tan H, Ashkan K, et al. Human subthalamic nucleus–medial frontal cortex theta phase

coherence is involved in conflict and error related cortical monitoring. *NeuroImage* 2016;137:178–87. doi:10.1016/j.neuroimage.2016.05.031

14 Riesel A, Endrass T, Auerbach LA, et al. Overactive Performance Monitoring as an Endophenotype for Obsessive-Compulsive Disorder: Evidence From a Treatment Study. *Am J Psychiatry* 2015;172:665–73. doi:10.1176/appi.ajp.2014.14070886

15 Norman LJ, Carlisi CO, Christakou A, et al. Neural dysfunction during temporal discounting in paediatric Attention-Deficit/Hyperactivity Disorder and Obsessive-Compulsive Disorder. *Psychiatry Research: Neuroimaging* 2017;269:97–105. doi:10.1016/j.psychresns.2017.09.008

16 Voon V, Droux F, Morris L, et al. Decisional impulsivity and the associative-limbic subthalamic nucleus in obsessive-compulsive disorder: stimulation and connectivity. *Brain* 2017;140:442–56. doi:10.1093/brain/aww309

17 Tyagi H, Apergis-Schoute AM, Akram H, et al. A Randomized Trial Directly Comparing Ventral Capsule and Anteromedial Subthalamic Nucleus Stimulation in Obsessive-Compulsive Disorder: Clinical and Imaging Evidence for Dissociable Effects. *Biol Psychiatry* 2019;85:726–34. doi:10.1016/j.biopsych.2019.01.017

18 Li N, Baldermann JC, Kibleur A, et al. A unified connectomic target for deep brain stimulation in obsessive-compulsive disorder. *Nature Communications* 2020;11:3364. doi:10.1038/s41467-020-16734-3

VERSION 2 – REVIEW

REVIEWER	Herz , Damian University of Oxford
REVIEW RETURNED	19-May-2021

GENERAL COMMENTS	The authors have sufficiently addressed my comments.
--

REVIEWER	Turner, Dennis Duke University Department of Neurobiology, Neurosurgery, Biomed Engineering
REVIEW RETURNED	24-May-2021

GENERAL COMMENTS	The authors have detailed their revisions and have responded appropriately to the suggestions.
--